# Prognostic role of interim F-18 fluorodeoxyglucose positron emission tomography-computed tomography during chemoradiation therapy in patients with hypopharyngeal squamous cell carcinoma

Takamitsu Mase[1], Yutaka Toyomasu[1], Hajime Ishinaga[2], Yui Nanpei[1¤a], Tomoko Kawamura[1¤b], Akinori Takada[1], Yasutaka Ichikawa[1], Noriko Ii[1¤c], Tomoya Hirata[2], Kazuhiko Takeuchi[2¤d], Hajime Sakuma[1], Yoshihito Nomoto[1]*

1 Department of Radiology, Mie University Hospital, Tsu, Mie, Japan, 2 Department of Otorhinolaryngology-Head and Neck Surgery, Mie University Graduate School of Medicine, Tsu, Mie, Japan

¤a Department of Radiology, Mie Prefectural General Medical Center, Yokkaichi, Mie, Japan.
¤b Department of Radiology, Matsusaka Chuo General Hospital, Matsusaka, Mie, Japan.
¤c Department of Radiation Oncology, Ise Red Cross Hospital, Ise, Mie, Japan.
¤d Department of Otorhinolaryngology, Matsusaka Chuo General Hospital, Matsusaka, Mie, Japan.
* nomoto-y@med.mie-u.ac.jp

## Abstract

This study aimed to investigate the usefulness of interim $^{18}$F fluorodeoxyglucose positron emission tomography–computed tomography (FDG PET/CT) during definitive radiation therapy (RT) as a prognostic indicator of disease recurrence in patients with hypopharyngeal squamous cell carcinoma. This prospective analysis included 35 patients with biopsy-proven hypopharyngeal squamous cell carcinoma who received platinum-based chemoradiotherapy and underwent pretreatment FDG PET/CT and interim FDG PET/CT (iPET) at a cumulative RT doses of 36.0–45.0 Gy. The maximum standardized uptake value (SUVmax), metabolic tumor volume, and total lesion glycolysis of the primary tumor (PT) and combined total lymph nodes for both pre-PET and iPET were analyzed, and their percentage reductions in iPET were calculated. The optimal cutoff values of the metabolic parameters were derived from receiver operating characteristic curves. The outcomes were compared between patients with metabolic parameters above and below the respective cutoff values. Disease recurrence (locoregional or distant) was defined as a biopsy-proven tumor or unequivocal clinical and radiological evidence of progression. Twelve (34%) patients experienced disease recurrence during a median follow-up of 52 months. Univariate Cox regression analysis revealed that the reduction ratio of the SUVmax of the PT (ΔSUVp; hazard ratio, 7.685; p = 0.008) was a significant predictor of disease recurrence. Kaplan–Meier curve analysis revealed that a smaller ΔSUVp was associated with worse progression-free survival (log-rank, p = 0.002). Metabolic parameters

**Data availability statement:** All relevant data are within the paper and its Supporting Information files.

**Funding:** The author(s) received no specific funding for this work.

**Competing interests:** The authors have declared that no competing interests exist.

measured using iPET may be useful predictors of disease recurrence in patients with hypopharyngeal squamous cell carcinoma treated with chemoradiotherapy. In this study, ΔSUVp was the best prognostic indicator.

## Introduction

Head and neck squamous cell carcinoma (HNSCC) represents a substantial global cancer burden and significantly contributes to cancer-related mortality worldwide [1]. Within this group, hypopharyngeal squamous cell carcinoma accounts for a relatively small subset of head and neck cancer [2]. It is often diagnosed at an advanced stage and has a poor prognosis, with 5-year survival rates ranging from 25% to 50%, depending on the stage and treatment [3].

Satisfactory results have been reported with the use of concomitant chemoradiotherapy for locally advanced hypopharyngeal cancer to preserve organ function or unresectable disease [4]. However, these treatment regimens are associated with severe late toxicities, such as pharyngeal/swallowing disturbances, which can have a lasting impact on a patient's quality of life [5]. Locoregional recurrence and distant metastasis are common, and the possibility of successful salvage treatment for recurrent disease is low. Therefore, new biological parameters should be identified to support risk stratification and individualization of treatments to minimize toxicity while maintaining therapeutic efficacy. The early identification of these parameters may improve patient outcomes.

[18]F fluorodeoxyglucose (FDG) positron emission tomography–computed tomography (PET/CT) is useful for staging and radiation therapy (RT) planning for head and neck cancers. FDG PET/CT has diagnostic value for the initial staging and detection of residual or recurrent disease [6]. Metabolic parameters, including the maximum standardized uptake value (SUVmax), metabolic tumor volume (MTV), and total lesion glycolysis (TLG), also offer predictive value for anticipating therapeutic response [7]. These parameters are prognostic indicators for patients with locally advanced HNSCC [8–13], and the prognostic value of FDG PET/CT during RT has been reported [14–17]. However, these reports pertain to a variety of head and neck sites, which may limit the generalizability of their data, as tumors from different sites behave differently from both clinical and biological perspectives. Therefore, independent evaluation of patients with hypopharyngeal cancer is warranted.

The aim of this study was to investigate the usefulness of interim FDG PET/CT (iPET), performed at a cumulative RT dose ranging from 36.0 to 45.0 Gy of primary RT, as a prognostic indicator of disease recurrence in patients with hypopharyngeal squamous cell carcinoma. In particular, we evaluated whether the residual metabolic tumor burden, determined via SUVmax, MTV, and TLG, correlated with disease recurrence and whether the metabolic response, assessed via the reduction ratio of these three metabolic parameters, also predicts disease recurrence.

## Materials and methods

### Study design

This was a prospective cohort study of patients who underwent iPET for hypopharyngeal cancer during CRT. All lesions were localized using flexible laryngoscopy, and the diagnosis of hypopharyngeal squamous cell carcinoma was confirmed histopathologically using biopsy specimens. The eligibility criteria were newly diagnosed and histologically confirmed hypopharyngeal squamous cell carcinoma. Patients with metastatic disease, other active malignancies, or uncontrolled diabetes mellitus were excluded.

### Procedures

The pretreatment and iPET images of patients with hypopharyngeal squamous cell carcinoma who received definitive RT with platinum-based chemotherapy were prospectively evaluated. RT was administered to the primary and neck regions once daily using 6-MV photons. A total irradiation dose of 70.2 Gy was delivered in daily fractions of 1.8 Gy over 8 weeks. After 41.4 Gy was administered, the clinical target volume was reduced to encompass only the primary region and involved neck nodes. RT was delivered using three-dimensional conformal radiation therapy (3D-CRT) or intensity-modulated radiation therapy (IMRT).

### Data collection and variables

The data from this cohort study were prospectively collected and managed. Clinical and demographic data were retrieved from institutional medical records and included age at diagnosis, sex, Eastern Cooperative Oncology Group (ECOG) performance status, smoking history, primary tumor (PT) subsite, T classification, N classification, and overall stage according to the 7th Union for International Cancer Control (UICC) TNM classification. For the purposes of this study, locally advanced disease was defined as stage III or IV hypopharyngeal squamous cell carcinoma according to the 7th UICC TNM classification, without radiologically evident distant metastasis. Specific chemotherapeutic regimens were administered. Imaging variables (SUVmax, MTV, and TLG) were derived from the pretreatment and iPET scans. The PET metrics were acquired and measured using the imaging software SYNAPSE SAI Viewer (version 2.0; FUJIFILM, Tokyo, Japan).

### PET/CT scanning technique

PET/CT was performed before treatment, and iPET scans were performed at cumulative RT doses of 36.0–45.0 Gy. The patients fasted for at least 6 h before the PET examinations, and their blood glucose levels were confirmed to be < 150 mg/dL. FDG PET/CT commenced 60 min after the injection of FDG (3.7 MBq/kg body weight) using the GE Discovery PET/CT 690 system (GE Healthcare, Milwaukee, WI, USA). The PET images covered the area from the skull to the midthigh. The head–neck PET image was acquired for 6 min per bed position in list mode. The PET images were reconstructed using three-dimensional ordered-subset expectation maximization (2 iterations, 24 subsets, and a 6-mm Gaussian filter) with time-of-flight and point-spread-function modeling and CT attenuation correction. The parameters of CT imaging for attenuation correction included a rotation time of 0.4 s, noise index of 12, tube voltage of 120 kV, and automatic exposure control (tube current range, 10–300 mA). Images were reconstructed to a 192 × 192 matrix with 3.27-mm slice thickness. All PET/CT studies were performed using the same PET/CT system with the same acquisition and reconstruction protocols.

### FDG PET image interpretation and metabolic parameter measurement

All FDG PET images were analyzed by two experienced radiation oncologists. PET metrics were acquired using SYNAPSE SAI Viewer version 2.0. The SUVmax was derived by selecting the area with the most avid uptake at the PT or nodal sites. The MTV was derived by applying a fixed standardized uptake value (SUV) threshold of 2.5 as the lowest limit of the segmentation criteria. TLG was calculated by multiplying the MTV by the mean standardized uptake value

(SUVmean). A fixed SUV threshold of 2.5 was chosen as the segmentation algorithm for the calculation of the MTV and TLG, consistent with other studies and meta-analyses [9,13,18]. The automatically derived, computer-assisted contours and regions of interest were verified on three cross-sectional images (axial, coronal, and sagittal) to ensure that the PT and nodal sites were accurately included, and adjacent normal structures were excluded. The SUVmax, SUVmean, MTV, and TLG were derived for the combined PT and total lymph nodes (PTN) from both pretreatment FDG PET/CT (prePET) and iPET. The reduction ratios of the SUVmax (ΔSUV), MTV (ΔMTV), and TLG (ΔTLG) between the two images were calculated. For example, ΔSUV was calculated using the following equation: $\Delta SUV = 1 - (\text{interim SUVmax/pretreatment SUVmax})$.

## Statistical analyses

The patients were observed for at least 2 years. Physical and endoscopic examinations were performed every 4–8 weeks for at least 2 years after treatment, and diagnostic imaging (CT and/or magnetic resonance imaging [MRI]) was performed every 3–6 months, when clinically indicated. Disease recurrence was defined as a biopsy-confirmed tumor whenever feasible. When biopsy was not possible, unequivocal progression on CT/MRI or FDG PET/CT, together with compatible clinical findings, was considered disease recurrence. No biochemical criteria were used. The primary endpoint was progression-free survival (PFS). The secondary endpoint was overall survival (OS). The follow-up duration was calculated from the first day of RT to the date of death or last follow-up visit. PFS was defined as the time from the date of the first RT to the date of death or locoregional or distant recurrence. Patients without progression or death were censored at the date of their last clinical or imaging follow-up. OS was defined as the time from the date of the first RT to the date of death from any cause, and patients who were alive were censored at the date of the last follow-up. Locoregional recurrences included recurrences in the original PT region and nodal recurrences within the region of external beam radiation fields. Distant metastases were defined as recurrences outside the external beam radiation field. A diagnosis of tumor recurrence or distant metastasis was based on either a positive biopsy result or unequivocal clinical or radiographic evidence of disease progression. PFS and OS were estimated using Kaplan–Meier curve analysis and compared using the log-rank test.

FDG PET/CT parameters were compared between the disease recurrence and non-disease recurrence groups using the Mann–Whitney U test. Receiver operating characteristic (ROC) curve analysis was performed to determine the effectiveness of the parameters in predicting disease recurrence. The areas under the ROC curves (AUCs) were calculated, and the optimal cutoff values were determined as the threshold that maximized the Youden index. For univariate analysis, Cox regression analysis was used to identify the prognostic factors for PFS and OS. To evaluate potential confounding, we summarized the baseline characteristics by the ΔSUVp cutoff (0.69) and tested the differences using the Mann–Whitney U test for continuous variables (age) and Fisher's exact test for categorical variables (sex, ECOG performance status [PS], smoking history, primary tumor subsite, T/N classification, stage, RT technique [IMRT vs. 3D-CRT], chemotherapeutic regimen) between two groups. Interobserver reliability for the measurement of FDG PET/CT parameters was assessed using the intraclass correlation coefficient (ICC). A two-sided p-value <0.05 was considered statistically significant. All analyses were performed using SPSS Statistics for Windows version 23.0 (IBM Corp., Armonk, NY, USA).

## Ethics statement

Participants were prospectively recruited between October 10, 2013, and March 31, 2020, at Mie University Hospital. All participants provided written informed consent before undergoing any study-specific procedures. The study protocol, consent documents, and all amendments were approved by the Ethics Review Committee for Medical Research of Mie University Hospital (permit no. 2615), which covered the entire recruitment period. Patients with minor errors were excluded.

## Results

### Patient characteristics

A total of 35 patients were included in the study from October 10, 2013, to March 31, 2020. The median age at diagnosis was 65 years (range, 46–81 years), and 33 patients (94%) were men. Most (97%) patients presented with locally advanced disease. Patients with T3 and T4 disease comprised 51% of the study population, whereas 69% had N2 or N3 disease. The most common subsite was the pyriform sinus, which accounted for 69% of the study population. RT was completed in all 35 patients. The patient characteristics are summarized in Table 1.

### Association between PET parameters and disease recurrence

The median metabolic values of the PT for prePET vs. iPET were as follows: SUVmax, 14.99 (range, 3.19–25.99) vs. 4.31 (1.55–10.90); MTV, 13.45 (0.20–45.80) vs. 2.65 (0–23.25) cm$^3$; TLG, 78.60 (0.70–328.90) vs. 9.3 (0–110.35). The median metabolic values of the index node for prePET vs. iPET were as follows: SUVmax, 7.65 (range, 0–29.32) vs. 2.64 (0–12.64); MTV, 1.60 (0–89.05) vs. 0.10 (0–85.20) cm$^3$; TLG, 6.60 (0–685.55) vs. 0.20 (0–525.05).

At the time of analysis, 26 patients (74%) were alive and 23 (66%) were disease-free, whereas 12 (34%) had treatment failure (9 with locoregional failure, 7 with distant failure, and 4 with both locoregional and distant failure). The median follow-up period of all patients was 52 months (range, 4–64 months; mean, 46 months), whereas that of survivors was 61 months (range, 30–64 months; mean, 55 months). Overall, 9 of the 35 patients (26%) died during the follow-up period: 7 from cancer and 2 from other causes. All recurrence-free patients were followed up for at least 2 years. After chemoradiotherapy, 12 patients exhibited disease recurrence, whereas 23 did not. Table 2 summarizes the association between PET parameters and disease recurrence. Significant differences were observed in the residual metabolic burden of iPET and the reduction ratios between the disease and non-disease recurrence groups, whereas no significant differences were observed in the prePET parameters.

### ROC analysis and cutoff values

The optimal cutoff values derived from the ROC analysis are summarized in Table 3. The cutoff values of PT for iPET were as follows: SUVmax, 4.85 (AUC = 0.80; p = 0.005); MTV, 4.05 cm$^3$ (AUC = 0.73; p = 0.029); TLG, 12.35 (AUC = 0.73; p = 0.029). The cutoff values of the PTN for iPET were as follows: MTV, 10.30 cm$^3$ (AUC = 0.75; p = 0.017); TLG, 27.75 (AUC = 0.76; p = 0.012). The cutoff values of the ΔSUVmax of the PT (ΔSUVp) were as follows: ΔSUVmax, 0.69 (AUC = 0.81; p = 0.003); ΔMTV, 0.77 (AUC = 0.73; p = 0.026); ΔTLG, 0.84 (AUC = 0.75; p = 0.018). The cutoff values of the ΔMTV of the PTN were as follows: ΔMTV, 0.77 (AUC = 0.76; p = 0.012); ΔTLG, 0.86 (AUC = 0.79; p = 0.006).

### Prognostic value of PET parameters

Table 4 summarizes the results of the univariate analysis. The residual metabolic burdens on iPET and the reduction ratio were significantly associated with disease recurrence. ΔSUVp > 0.69 was associated with a favorable PFS (hazard ratio [HR], 7.685; 95% confidence interval [CI], 1.706–34.626; p = 0.008). An interim SUVmax of the PT ≤ 4.85 was associated with a favorable OS (HR, 12.366; 95% CI, 1.525–100.3; p = 0.019). Kaplan–Meier curve analysis (Fig 1) demonstrated that patients with ΔSUVp ≤ 0.69 had significantly lower rates of PFS (3-year rate, 37% vs. 94%; p = 0.002; Fig 1A) than those with ΔSUVp > 0.69. ΔSUVp ≤ 0.69 showed a trend toward worse OS (62% vs. 94%; p = 0.064; Fig 1B) than ΔSUVp > 0.69.

The baseline characteristics stratified by the ΔSUVp groups (≤0.69 vs. >0.69) are presented in Table 5; no statistically significant between-group differences were observed across age, sex, ECOG performance status, smoking history, PT subsite, T/N classification, stage, RT technique (IMRT vs. 3D-CRT) or chemotherapeutic regimen (all p > 0.05).

**Table 1. Patient characteristics (n = 35).**

| | |
|---|---|
| **Follow-up time (months)** | **52 [4–64]** |
| **Age (years)** | 65 [46–81] |
| **Sex** | |
| Male | 33 (94) |
| Female | 2 (6) |
| **ECOG PS** | |
| 0 | 19 (54) |
| 1 | 14 (40) |
| 2 | 2 (6) |
| **Smoking history** | |
| Smoker | 33 (94) |
| Non-smoker | 2 (6) |
| **Primary tumor subsite** | |
| Pyriform sinus | 24 (69) |
| Posterior pharyngeal wall | 7 (20) |
| Postcricoid region | 4 (11) |
| **T classification** | |
| 1 | 2 (6) |
| 2 | 15 (43) |
| 3 | 10 (29) |
| 4 | 8 (23) |
| **N classification** | |
| 0 | 3 (9) |
| 1 | 8 (23) |
| 2 | 22 (63) |
| 3 | 2 (6) |
| **Stage (7th UICC TNM classification)** | |
| II | 1 (3) |
| III | 8 (23) |
| IV | 26 (74) |
| **Radiation dose: 70.2 Gy in 39 fractions** | 35 (100) |
| **RT technique** | |
| IMRT | 23 (66) |
| Conventional 3D-CRT | 12 (34) |
| **Chemotherapeutic regimen** | |
| 5-Fluorouracil + cisplatin | 14 (40) |
| Cisplatin alone | 14 (40) |
| 5-Fluorouracil + carboplatin | 4 (11) |
| Carboplatin alone | 3 (9) |

Values are presented a median [range] or n (%).

ECOG PS, Eastern Cooperative Oncology Group performance status; UICC, Union for International Cancer Control; TNM, tumor, node, metastasis; RT, radiation therapy; IMRT, intensity-modulated radiation therapy; 3D-CRT, three-dimensional conformal radiation therapy.

**Table 2. Association between PET parameters and disease recurrence for the 35 patients.**

| Parameter | Disease recurrence (+) (n = 12) | Disease recurrence (−) (n = 23) | p-Value |
|---|---|---|---|
| **Pretreatment values of each parameter** | | | |
| **preSUVpmax** | 14.1 (9.7–17.9) | 15.2 (11.4–18.3) | 0.851 |
| **preMTVp (cm³)** | 18.6 (6.7–28.4) | 12.7 (5.4–28.6) | 0.572 |
| **preTLGp** | 121.5 (28.8–186.9) | 68.9 (29.7–164.9) | 0.668 |
| **preMTVptn (cm³)** | 36.1 (11.7–43.8) | 15.3 (8.9–34.8) | 0.161 |
| **preTLGptn** | 214.4 (65.0–293.7) | 84.5 (41.7–201.1) | 0.172 |
| **Interim values of each parameter** | | | |
| **iSUVpmax** | 6.1 (4.9–8.0) | 4.0 (2.6–5.1) | 0.004 |
| **iMTVp (cm³)** | 4.9 (1.6–8.3) | 2.0 (0.1–4.5) | 0.028 |
| **iTLGp** | 17.8 (5.8–35.8) | 6.1 (0.3–16.6) | 0.028 |
| **iMTVptn (cm³)** | 8.3 (3.4–13.6) | 2.3 (0.2–6.3) | 0.016 |
| **iTLGptn** | 30.3 (13.4–55.0) | 9.1 (0.6–20.2) | 0.011 |
| **Delta values of each parameter** | | | |
| **ΔSUVp** | 0.556 (0.443–0.670) | 0.750 (0.641–0.807) | 0.002 |
| **ΔMTVp** | 0.644 (0.557–0.848) | 0.862 (0.717–0.981) | 0.026 |
| **ΔTLGp** | 0.805 (0.712–0.907) | 0.930 (0.851–0.988) | 0.017 |
| **ΔMTVptn** | 0.720 (0.607–0.767) | 0.853 (0.745–0.965) | 0.011 |
| **ΔTLGptn** | 0.817 (0.732–0.861) | 0.909 (0.856–0.981) | 0.005 |

Values are presented a median [interquartile range].

PET, positron emission tomography; IQR, interquartile range; SUVpmax, maximum standardized uptake value of the primary tumor; MTV, metabolic tumor volume; TLG, total lesion glycolysis; preSUVpmax, pretreatment SUVpmax; preMTVp, pretreatment MTV of the primary tumor; preTLGp, pretreatment TLG of the primary tumor; preMTVptn, pretreatment MTV of the combined primary tumor and total lymph nodes; preTLGptn, pretreatment TLG of the combined primary tumor and total lymph nodes; iSUVpmax, interim SUVpmax; iMTVp, interim MTV of primary tumor; iTLGp, interim TLG of primary tumor; iMTVptn, interim MTV of the combined primary tumor and total lymph nodes; iTLGptn, interim TLG of the combined primary tumor and total lymph nodes; ΔSUVp, reduction ratio of SUVpmax; ΔMTVp, reduction ratio of MTV of the primary tumor; ΔTLGp, reduction ratio of TLG of the primary tumor; ΔMTVptn, reduction ratio of MTV of the combined primary tumor and total lymph nodes; ΔTLGptn, reduction ratio of TLG of the combined primary tumor and total lymph nodes.

## High interobserver reliability

High interobserver reliability was observed for all parameters. The ICCs of the PT and PTN for pre-PET and iPET are presented in Table 6.

## Discussion

To the best of our knowledge, this is the first prospective study to evaluate the prognostic value of FDG-PET/CT before and during RT for hypopharyngeal cancer. This report suggests that pre-PET and iPET parameters may be useful prognostic factors for patients with hypopharyngeal cancer. These metabolic parameters have the potential to stratify patients according to prognosis (poor and good). Therefore, iPET parameters may allow for the relatively early detection of treatment-resistant diseases that are not detected via pre-PET alone, allowing for individually adapted treatment.

Data on the prognostic value of the residual metabolic burden of iPET in HNSCC are limited. In a study by Farrag et al., the SUVmax of PTs at baseline and during treatment after 47 Gy was associated with OS [19]. Kim et al. reported that the TLG of the PT during RT was the most statistically significant prognostic factor for OS and PFS [17]. In this study, the associations between the reduction ratios of metabolic values and oncological outcomes were not well studied. Hentschel et al. evaluated iPET in patients with HNSCC and demonstrated that a decrease of 50% or more in SUVmax of the PT from the beginning (0 Gy) to 1 week or 2 weeks (10 or 20 Gy) of treatment is a potential prognostic marker for patients

**Table 3. Cutoff values for the prediction of treatment outcomes using receiver operating characteristic analysis.**

| Parameter | Cutoff value | AUC (95% CI) | Sensitivity | Specificity | p-Value |
|---|---|---|---|---|---|
| **Interim values of each parameter** | | | | | |
| **iSUVpmax** | 4.85 | 0.80 (0.63–0.95) | 0.83 | 0.70 | 0.005 |
| **iMTVp (cm³)** | 4.05 | 0.73 (0.55–0.90) | 0.67 | 0.74 | 0.029 |
| **iTLGp** | 12.35 | 0.73 (0.55–0.90) | 0.75 | 0.70 | 0.029 |
| **iMTVptn (cm³)** | 10.30 | 0.75 (0.57–0.93) | 0.50 | 0.96 | 0.017 |
| **iTLGptn** | 27.75 | 0.76 (0.59–0.94) | 0.58 | 0.91 | 0.012 |
| **Delta values of each parameter** | | | | | |
| **ΔSUVp** | 0.69 | 0.81 (0.67–0.95) | 0.92 | 0.65 | 0.003 |
| **ΔMTVp** | 0.77 | 0.73 (0.56–0.91) | 0.75 | 0.74 | 0.026 |
| **ΔTLGp** | 0.84 | 0.75 (0.58–0.91) | 0.67 | 0.78 | 0.018 |
| **ΔMTVptn** | 0.77 | 0.76 (0.59–0.93) | 0.83 | 0.74 | 0.012 |
| **ΔTLGptn** | 0.86 | 0.79 (0.62–0.95) | 0.83 | 0.74 | 0.006 |

AUC, area under the receiver operating characteristic curve; CI, confidence interval; SUVpmax, maximum standardized uptake value of the primary tumor; MTV, metabolic tumor volume; TLG, total lesion glycolysis; iSUVpmax, interim SUVpmax; iMTVp, interim MTV of the primary tumor; iTLGp, interim TLG of the primary tumor; iMTVptn, interim MTV of the combined primary tumor and total lymph nodes; iTLGptn, interim TLG of the combined primary tumor and total lymph nodes; ΔSUVp, reduction ratio of SUVpmax; ΔMTVp, reduction ratio of MTV of primary tumor; ΔTLGp, reduction ratio of TLG of the primary tumor; ΔMTVptn, reduction ratio of MTV of the combined primary tumor and total lymph nodes; ΔTLGptn, reduction ratio of TLG of the combined primary tumor and total lymph nodes.

**Table 4. Univariate analysis for progression-free and overall survival for all patients.**

| Parameter | Progression-free survival | | | Overall survival | | |
|---|---|---|---|---|---|---|
| | HR | 95% CI | p-Value | HR | 95% CI | p-Value |
| **Interim values of each parameter** | | | | | | |
| **ISUVpmax (≥4.85 vs. <4.85)** | 6.16 | 1.70–22.42 | 0.006 | 12.37 | 1.53–100.3 | 0.019 |
| **iMTVp (≥4.05 vs. <4.05 cm³)** | 2.91 | 1.00–8.44 | 0.05 | 2.39 | 0.64–8.91 | 0.196 |
| **ITLGp (≥12.35 vs. <12.35)** | 3.29 | 1.09–9.91 | 0.034 | 3.21 | 0.80–12.94 | 0.101 |
| **iMTVptn (≥10.30 vs. <10.30 cm³)** | 6.92 | 2.26–21.21 | 0.001 | 10.53 | 2.68–41.42 | 0.001 |
| **iTLGptn (≥27.75 vs. <27.75)** | 5.35 | 1.80–15.86 | 0.003 | 6.38 | 1.64–24.87 | 0.008 |
| **Delta values of each parameter** | | | | | | |
| **ΔSUVp (≤0.69 vs. >0.69)** | 7.69 | 1.71–34.63 | 0.008 | 3.97 | 0.82–19.30 | 0.087 |
| **ΔMTVp (≤0.77 vs. >0.77)** | 3.68 | 1.23–11.07 | 0.02 | 6.53 | 1.35–31.70 | 0.02 |
| **ΔTLGp (≤0.84 vs. >0.84)** | 3.29 | 1.13–9.55 | 0.029 | 8.81 | 1.81–42.87 | 0.007 |
| **ΔMTVptn (≤0.77 vs. >0.77)** | 4.58 | 1.43–14.72 | 0.011 | 5.78 | 1.19–28.04 | 0.03 |
| **ΔTLGptn (≤0.86 vs. >0.86)** | 4.58 | 1.43–14.72 | 0.011 | 5.78 | 1.19–28.04 | 0.03 |

HR, hazard ratio; CI, confidence interval; SUVpmax, maximum standardized uptake value of the primary tumor; MTV, metabolic tumor volume; TLG, total lesion glycolysis; iSUVpmax, interim SUVpmax; iMTVp, interim MTV of the primary tumor; iTLGp, interim TLG of the primary tumor; iMTVptn, interim MTV of the combined primary tumor and total lymph nodes; iTLGptn, interim TLG of the combined primary tumor and total lymph nodes; ΔSUVp, reduction ratio of SUVpmax; ΔMTVp, reduction ratio of MTV of the primary tumor; ΔTLGp, the reduction ratio of the TLG of the primary tumor; ΔMTVptn, reduction ratio of the MTV of the combined primary tumor and total lymph nodes; ΔTLGptn, reduction ratio of TLG of the combined primary tumor and total lymph nodes.

with HNSCC [20]. Chen et al. studied the SUVp after a cumulative dose of 40–50 Gy during RT and reported on a correlation between the ΔSUVp and oncological outcomes [14]. Similarly, more recent studies by Martens et al. [21] and Kim et al. [22] further support the utility of interim PET parameters, both residual metabolic activity and its reduction rate, for

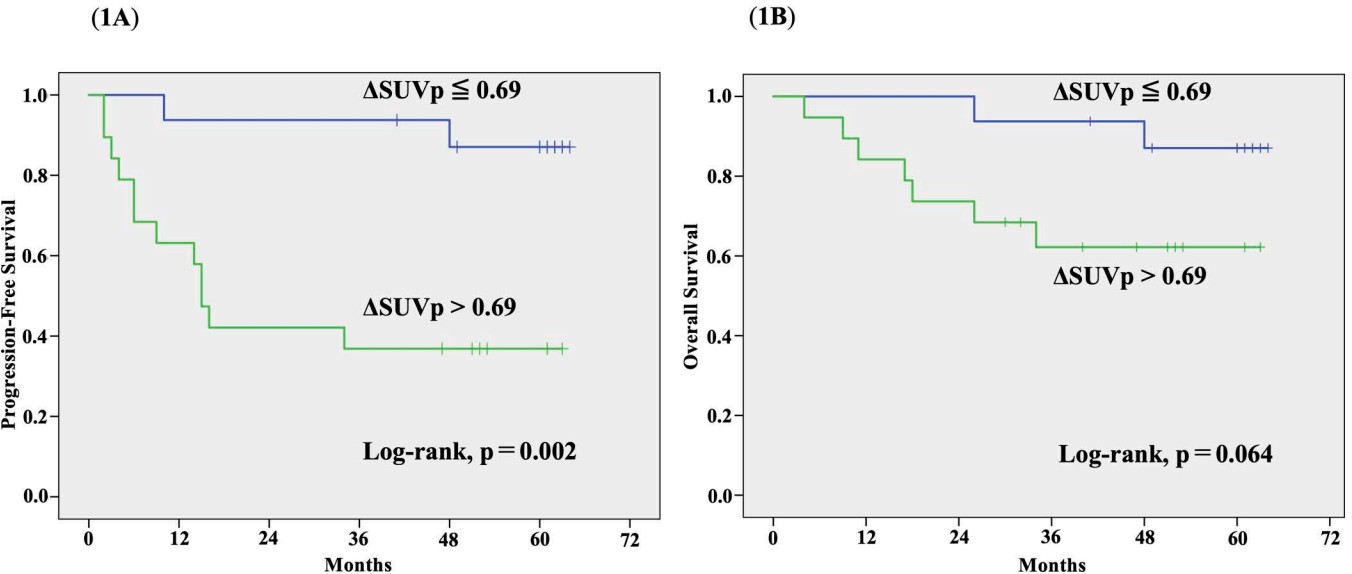

**Fig 1. Kaplan–Meier curves stratified by ΔSUVp cutoff (0.69): (A) progression-free survival by ΔSUVp and (B) overall survival by ΔSUVp.** Patients with a ΔSUVp ≤ 0.69 had significantly worse PFS than those with a ΔSUVp > 0.69 (p = 0.002). ΔSUVp, SUVmax of the primary tumor.

outcome stratification in HNSCC patients undergoing RT. However, Min et al. evaluated iPET in the third week of RT and discovered no significant associations between the percentage reduction of any of the metabolic parameters (ΔSUVmax, ΔMTV, and ΔTLG) and oncological outcomes [18].

In the present study, which only included patients with hypopharyngeal squamous cell carcinoma, the most notable finding was that the ΔSUVp obtained on iPET provided better prognostic stratification than the residual metabolic burden itself. This observation is generally consistent with previous reports on iPET in heterogeneous HNSCC cohorts, in which both the residual metabolic activity and its percentage reduction were associated with treatment outcomes [16,17,21,22]. However, our data specifically refined these findings for hypopharyngeal cancer. In this subsite, the degree of metabolic response during chemoradiotherapy may be a more relevant prognostic indicator than the absolute metabolic activity at the interim time point. A plausible explanation is that hypopharyngeal squamous cell carcinoma frequently presents with locally advanced disease and an intrinsically poorer prognosis than other head and neck subsites [23]; thus, inter-patient differences in how rapidly the PT shows metabolic regression during treatment may better capture treatment resistance. These results support the need to evaluate hypopharyngeal squamous cell carcinoma separately from mixed HNSCC cohorts and indicate that subsite-specific cutoff values for iPET-derived parameters should be considered when iPET is used for early risk stratification. Recent head and neck oncology reports have proposed several subsite-specific treatments or prognostic strategies, particularly for nasopharyngeal carcinoma, indicating a move toward tailoring radiotherapy and risk stratification to each subsite's biological and anatomical features [24,25]. These studies suggest that HNSCC should not be considered a homogeneous disease. Against this background, our study adds subsite-specific evidence for hypopharyngeal squamous cell carcinoma by showing that an iPET-derived metabolic response during chemoradiotherapy can help identify patients at a higher risk of progression in this unfavorable subsite.

In addition to this prognostic significance, the use of the reduction ratio, rather than reliance on an absolute value (the residual metabolic burden), also has the methodological benefit of improving the standardization and reproducibility of longitudinal studies. This benefit is attributable to the lower susceptibility of this parameter to treatment-related confounding

**Table 5. Comparison of patient characteristics according to the ΔSUVp cutoff value.**

| Characteristics | Total (n = 35) | ΔSUVp ≤ 0.69 (n = 19) | ΔSUVp > 0.69 (n = 16) | p-Value |
|---|---|---|---|---|
| Age (years) | 65 [46–81] | 65 [46–74] | 68 [47–81] | 0.168 |
| Sex | | | | 0.202 |
| Male | 33 (94) | 19 (100) | 14 (88) | |
| Female | 2 (6) | 0 (0) | 2 (12) | |
| ECOG PS | | | | 0.500 |
| 0 | 19 (54) | 9 (47) | 10 (62) | |
| 1–2 | 16 (46) | 10 (53) | 6 (38) | |
| Smoking history | | | | 0.545 |
| Smoker | 33 (94) | 17 (89) | 16 (100) | |
| Non-smoker | 2 (6) | 2 (11) | 0 (0) | |
| Primary tumor subsite | | | | 0.974 |
| Pyriform sinus | 24 (69) | 13 (68) | 11 (69) | |
| Posterior pharyngeal wall | 7 (20) | 4 (21) | 3 (19) | |
| Postcricoid region | 4 (11) | 2 (11) | 2 (12) | |
| T classification | | | | 0.505 |
| T1–2 | 17 (49) | 8 (42) | 9 (56) | |
| T3–4 | 18 (51) | 11 (58) | 7 (44) | |
| N classification | | | | 1.000 |
| N0–1 | 11 (31) | 6 (32) | 5 (31) | |
| N2–3 | 24 (69) | 13 (68) | 11 (69) | |
| Stage (7th UICC TNM classification) | | | | 0.273 |
| II-III | 11 (31) | 4 (21) | 7 (44) | |
| IV | 24 (69) | 15 (79) | 9 (56) | |
| RT technique | | | | 1.000 |
| IMRT | 23 (66) | 12 (63) | 11 (69) | |
| Conventional 3D-CRT | 12 (34) | 7 (37) | 5 (31) | |
| Chemotherapeutic regimen | | | | 0.677 |
| Cisplatin-based | 28 (80) | 16 (84) | 12 (75) | |
| Non-cisplatin-based | 7 (20) | 3 (16) | 4 (25) | |

Values are presented a median [range] or n (%).

SUVpmax, maximum standardized uptake value of the primary tumor; ΔSUVp, reduction ratio of SUVpmax; ECOG PS, Eastern Cooperative Oncology Group performance status; UICC, Union for International Cancer Control; TNM, tumor; nodes; metastases; RT, radiation therapy; IMRT, intensity-modulated radiation therapy; 3D-CRT, three-dimensional conformal radiation therapy.

factors, such as mucositis, and differences in equipment and protocols among hospitals. Therefore, we expect that the selection of optimal ΔSUVp values will minimize inherent and individual site biases.

Our results imply that the ΔSUVp of patients with advanced hypopharyngeal cancer is useful for the identification of patients at high risk of disease recurrence. Patients with a low ΔSUVp may require a more aggressive treatment plan and shorter follow-up intervals after treatment. Further research is warranted to determine the optimal timing of iPET. Although earlier iPET may be useful in distinguishing metabolic from inflammatory changes and aiding early decisions concerning treatment-plan modifications [16], later assessments may also be valid.

Several reports have indicated that dose escalation may improve outcomes in head and neck cancer [26–28]; however, therapy-related complications may be increased by dose escalation because of the high radiation dose delivered to normal tissues [29]. Therefore, such escalation should be performed by irradiating a small field with a cone-down boost.

**Table 6. Intraclass correlation coefficients.**

| Parameter | ICCs (95% CI) |
|---|---|
| **Pretreatment values of each parameter** | |
| **preSUVpmax** | 1.000 (1.000–1.000) |
| **preMTVp (cm³)** | 0.999 (0.997–0.999) |
| **preTLGp** | 1.000 (1.000–1.000) |
| **preMTVptn (cm³)** | 1.000 (0.999–1.000) |
| **preTLGptn** | 1.000 (1.000–1.000) |
| **Interim values of each parameter** | |
| **iSUVpmax** | 1.000 (1.000–1.000) |
| **iMTVp (cm³)** | 1.000 (0.999–1.000) |
| **iTLGp** | 1.000 (0.999–1.000) |
| **iMTVptn (cm³)** | 1.000 (1.000–1.000) |
| **ITLGptn** | 1.000 (1.000–1.000) |

ICCs, intraclass correlation coefficients; CI, confidence interval; SUVpmax, maximum standardized uptake value of the primary tumor; MTV, metabolic tumor volume; TLG, total lesion glycolysis; preSUVpmax, pretreatment SUVpmax; preMTVp, pretreatment MTV of the primary tumor; preTLGp, pretreatment TLG of the primary tumor; preMTVptn, pretreatment MTV of the combined primary tumor and total lymph nodes; preTLGptn, pretreatment TLG of the combined primary tumor and total lymph nodes; iSUVpmax, interim SUVpmax; iMTVp, interim MTV of the primary tumor; iTLGp, interim TLG of the primary tumor; iMTVptn, interim MTV of the combined primary tumor and total lymph nodes; iTLGptn, interim TLG of the combined primary tumor and total lymph nodes.

In the current study, we confirmed that the MTV was sufficiently reduced at the time of iPET performed at a cumulative RT dose of 36–45 Gy. Therefore, the volume of normal tissue that receives high doses may be reduced, and the treatment effect may be improved by optimizing the timing of iPET and considering dose escalation.

This study has notable strengths, including its prospective design and relatively homogeneous patient population. Another strength of this study is that it evaluated only with patients with hypopharyngeal cancer treated with definitive chemoradiotherapy; this is important, as tumors from different sites behave differently from clinical and biological perspectives [23]. In accordance with the consensus recommendations for the use of FDG PET [30], all our patients underwent pretreatment and interim imaging via the same FDG PET protocol and were analyzed using the same software to minimize variability.

However, this study has limitations. A primary limitation of this study is its small sample size (n = 35) from a single center, which reduced the statistical power and resulted in only 12 disease recurrences, making it impossible to perform a robust multivariable Cox analysis to adjust for potential confounders such as tumor stage or chemotherapeutic regimens. Although our univariate analyses identified significant associations (Table 4), baseline tumor and treatment factors did not differ between the ΔSUVp-defined groups (Table 5), making a major confounding effect less likely.

In addition, the radiation oncologists who analyzed the FDG PET/CT scans were not blinded to the clinical context or outcomes, which could introduce potential interpretation bias. Although the treating physicians were not blinded to the iPET findings, these findings (including ΔSUVp) were not used to make real-time adaptive treatment decisions. Therefore, the observed prognostic value was unlikely to be confounded by iPET-based treatment modifications.

Another limitation is the method of cutoff determination. In clinical scenarios that prioritize early detection of treatment-resistant diseases, a more sensitivity-oriented cutoff than the statistically balanced cutoff may be preferable. However, in this study, we used a ROC-based cutoff derived from a small cohort (n = 35), which was susceptible to overfitting. Owing to the limited sample size, we were unable to perform meaningful internal validation (e.g., cross-validation

or split-sample testing). Therefore, the ΔSUVp cutoff of 0.69, which was derived using the Youden index, should be considered exploratory and intended for hypothesis generation. This finding needs to be confirmed in larger, independent cohorts before it can be considered for clinical use. Finally, although our findings identify a high-risk group, the clinical utility of this ΔSUVp threshold to guide adaptive therapy, such as dose escalation or regimen modification, remains speculative and unproven. This issue should be addressed in future prospective interventional trials.

## Conclusions

Residual volumetric metabolic burden and percentage reduction derived from iPET appear to be useful imaging biomarkers for predicting the risk of disease recurrence in patients with hypopharyngeal cancer treated with chemoradiotherapy. These biomarkers may play a role in individualized adaptive chemoradiotherapy, particularly with respect to the selection of dose intensification and adjuvant chemotherapy.

## Supporting information

**S1 File. Supplementary file.**
(XLSX)

## Acknowledgments

We would like to thank Editage (www.editage.jp) for the English language editing.

## Author contributions

**Conceptualization:** Takamitsu Mase, Yutaka Toyomasu, Yoshihito Nomoto.

**Data curation:** Takamitsu Mase, Yutaka Toyomasu, Hajime Ishinaga, Yui Nanpei, Tomoko Kawamura, Akinori Takada, Tomoya Hirata.

**Formal analysis:** Takamitsu Mase, Yutaka Toyomasu.

**Investigation:** Takamitsu Mase, Hajime Ishinaga, Yui Nanpei, Tomoko Kawamura, Akinori Takada, Tomoya Hirata.

**Methodology:** Yasutaka Ichikawa, Noriko Ii, Yoshihito Nomoto.

**Resources:** Hajime Ishinaga, Yui Nanpei, Tomoko Kawamura, Akinori Takada, Tomoya Hirata.

**Supervision:** Yasutaka Ichikawa, Noriko Ii, Kazuhiko Takeuchi, Hajime Sakuma, Yoshihito Nomoto.

**Validation:** Akinori Takada, Kazuhiko Takeuchi, Hajime Sakuma, Yoshihito Nomoto.

**Visualization:** Yutaka Toyomasu.

**Writing – original draft:** Takamitsu Mase, Yutaka Toyomasu, Yasutaka Ichikawa, Noriko Ii, Yoshihito Nomoto.

**Writing – review & editing:** Kazuhiko Takeuchi, Hajime Sakuma, Yoshihito Nomoto.

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
