## [Decision Letter · Decision Letter 0]

20 Oct 2025

Dear Dr. Nomoto,

Thank you for submitting your manuscript to PLOS ONE. After careful consideration, we feel that it has merit but does not fully meet PLOS ONE’s publication criteria as it currently stands. Therefore, we invite you to submit a revised version of the manuscript that addresses the points raised during the review process.

**ACADEMIC EDITOR:**

We look forward to receiving your revised manuscript.

Kind regards,

Carmelo Caldarella, Ph.D., M.D.

Academic Editor

PLOS ONE

Journal Requirements:

Additional Editor Comments:

Dear Authors, the manuscript has been evaluated in details by three Reviewers.

Based on their comments and suggestions, you should revise the manuscript, particularly specifying better the statistical analysis for cut-off calculation and Cox-regression, as well as better explain methods used (reference standard to confirm SCC, PET segmentation and follow-up duration).

Waiting for the revised version of the manuscript

Best regards.

Reviewers' comments:

Reviewer's Responses to Questions

**Comments to the Author**

1. Is the manuscript technically sound, and do the data support the conclusions?

Reviewer #1: Partly

Reviewer #2: Yes

Reviewer #3: Yes

2. Has the statistical analysis been performed appropriately and rigorously?

Reviewer #1: I Don't Know

Reviewer #2: Yes

Reviewer #3: Yes

3. Have the authors made all data underlying the findings in their manuscript fully available?

Reviewer #1: No

Reviewer #2: No

Reviewer #3: Yes

4. Is the manuscript presented in an intelligible fashion and written in standard English?

Reviewer #1: Yes

Reviewer #2: Yes

Reviewer #3: Yes

Reviewer #1: The study prospectively evaluated 35 patients with hypopharyngeal squamous cell carcinoma who underwent pretreatment and interim F-18 FDG PET/CT during chemoradiation to explore prognostic metabolic indicators. The authors analyzed SUVmax, metabolic tumor volume (MTV), and total lesion glycolysis (TLG) of the primary tumor and lymph nodes before and during treatment. They found that the reduction ratio of the primary tumor’s SUVmax (ΔSUVp) during radiotherapy was the most significant predictor of disease recurrence (HR = 7.685, p = 0.008), with smaller reductions associated with poorer progression-free survival. The authors conclude that interim PET/CT-derived metabolic changes, particularly ΔSUVp, may serve as useful prognostic biomarkers to guide individualized chemoradiation strategies in hypopharyngeal cancer.

I have following major concerns: 1) The sample size (n = 35) is too small for reliable statistical inference, and no multivariate analysis was conducted to adjust for potential confounders such as tumor stage or treatment regimen. 2) The heterogeneity in chemotherapy protocols (cisplatin alone vs. combination regimens) could introduce bias in metabolic response and outcomes. 3) ROC-derived cutoffs may be overfitted due to the small dataset and lack of cross-validation. 4) In the introduction or discussion part, following related references could be added to strengthen the paper: PMID: 40336569; doi.org/10.1002/pro6.70007. 5) The statistical analysis relied solely on univariate Cox regression, which may exaggerate the predictive value of ΔSUVp. 6) The clinical implications are limited, as how the identified thresholds would guide treatment adaptation remains speculative. 7) The methodology for PET segmentation (fixed SUV = 2.5) may not be optimal for all lesions, potentially affecting MTV/TLG quantification. Could the authors give us an explanation? 8) No discussion was provided on interscan variability, attenuation correction, or scanner calibration, which could influence SUV reproducibility. 9) Figures and tables could be improved for clarity, with overlapping abbreviations and dense data presentation making interpretation difficult. 10) Finally, the follow-up duration and censoring information are not clearly detailed for survival analyses, raising concerns about data completeness and robustness.

Reviewer #2: It was a pleasure to review your paper on a prospective study on the utilization of interim FDG PET/CT scanning during the chemoRT of patients with specifically hypopharyngeal cancer as a prognostic indicator. It has good data, decent design, and aligns with similar research in the field, although its more specific nature will assuredly give more credence to this implementation in this specific type of cancer. There are many things that I think should be addressed before acceptance. I have attached a PDF with comments and I'll summarize the most important points here:

1. There is a lack of citing more recent relevant studies which are extremely similar and analogous to this work including: https://doi.org/10.1016/j.oraloncology.2018.11.005 ; https://doi.org/10.1002/lary.24826 ; https://doi.org/10.1007/s00259-017-3836-8 ; and more found in this review: https://doi.org/10.1016/j.ijrobp.2017.02.217

2. There is a lack of an important table looking at your defined prognostic cutoff and the demographics to address confounders

3. The supplemental raw data is missing

4. There is a lack of explanation about the machines and software used. There is a lack of discussion on the limitations and about the lack of blinding, as well as how treatment may have been changed relative to the dSUVmax or not. These things should be explained.

5. You define a cutoff by maximizing Younden via the ROC, but this may not be the best approach, there should be discussion about this and the utility for prognostication for higher sensitivity or specificity.

I believe that these points can be addressed as Minor Revisions and will greatly improve the quality of the paper as it is Accepted.

Reviewer #3: In this manuscript authors aimed to assess the prognostic utility of interim FDG PET in patients with hypopharyngeal SCC undergoing chemo-radiotherapy. In my review, I’ve found several limitations and drawbacks mandating further improvement to enhance overall quality.

1. Abstract:

o At line 6: Phrase “hypopharyngeal SCC” → What was the reference standard for hypopharyngeal SCC? Was this biopsy-proven?

o At line 15: Phrase “Disease recurrence ” → Based on which metrics? Clinical, biochemical or by imaging?

2. Introduction:

o Please introduce: Baseline statistical facts about head and neck SCC in general and hypopharyngeal SCC in specific.

o Please introduce the role of FDG PET/CT in head and neck SCC. This include

- Diagnostic value (DOI: 10.2967/jnumed.124.268049)

- Predictive value (DOI: 10.3390/cancers15225461)

- Prognostic value (DOI: 10.3390/jcm12103514)Define FDA upon its first mention

3. Material and methods:

o Study design: Line 56 and 57 ( The eligibility criteria were newly diagnosed hypopharyngeal squamous cell carcinoma) → Please indicate the adopted reference standard to confirm hypopharyngeal SCC.

o Procedures: Please dedicate a new section to outline collected data variables and the software sued for data collection.

o PET/CT scanning technique: Please specify the cancer staging system utilized.

o Statistical analysis: Line 103 ( . The primary endpoint was progression-free survival (PFS). ) → Please accurately reflect and define the term progression. Was this based on radiographic, molecular imaging, clinical, or biochemical metrics?

4. Results:

o I advise the authors to apply relevant subheading/sub-labeling of paragraph shared in results section. Those that outline PET metrics can be put in sperate subheading from those detailing survival analysis.

o Line 136: (locally advanced disease) → The term locally advanced disease should be briefly explained in your methodology.

o Line 136-137: (T3 and T4 disease comprised 51% of the study population) → Please make sure to explain which cancer staging system was used for TNM categorization. Indicate this in our methos section.

o Line 174: (combined primary tumor and total lymph nodes) → For interim PET which response criteria was used to determine recurrence.

5. Discussion

o Add a concise discussion segment summarizing your study results in hypopharyngeal SCC comparing Prognostic role of interim F-18 fluorodeoxyglucose positron emission tomography computed tomography during chemoradiation therapy to other head and neck SCC in previous studies.

**Do you want your identity to be public for this peer review?** For information about this choice, including consent withdrawal, please see our Privacy Policy

Reviewer #1: No

Reviewer #2: **Yes:** Ryan J. Dikdan

Reviewer #3: **Yes:** Dhuha Ali Al-Adhami

---

## [Author Response · Author response to Decision Letter 1]

8 Dec 2025

Rebuttal Letter

We sincerely thank the editor and reviewers for their thoughtful and constructive comments on our manuscript. We have carefully revised the manuscript in line with these suggestions. Below, we present a point-by-point response to each comment. All corresponding revisions are highlighted in the updated manuscript.

Reviewer #1: The study prospectively evaluated 35 patients with hypopharyngeal squamous cell carcinoma who underwent pretreatment and interim F-18 FDG PET/CT during chemoradiation to explore prognostic metabolic indicators. The authors analyzed SUVmax, metabolic tumor volume (MTV), and total lesion glycolysis (TLG) of the primary tumor and lymph nodes before and during treatment. They found that the reduction ratio of the primary tumor’s SUVmax (ΔSUVp) during radiotherapy was the most significant predictor of disease recurrence (HR = 7.685, p = 0.008), with smaller reductions associated with poorer progression-free survival. The authors conclude that interim PET/CT-derived metabolic changes, particularly ΔSUVp, may serve as useful prognostic biomarkers to guide individualized chemoradiation strategies in hypopharyngeal cancer.

Response: We thank the reviewer for these positive comments on the work.

Issue 1-1: The sample size (n = 35) is too small for reliable statistical inference, and no multivariate analysis was conducted to adjust for potential confounders such as tumor stage or treatment regimen.

Response 1-1: Thank you for your insightful comment regarding the limitations of our study. We agree with the reviewer that the small sample size (n=35) is a primary limitation of this study. As the reviewer noted, our cohort was small and only 12 progression events occurred, so constructing a fully adjusted multivariable model for factors such as TNM stage or chemotherapy regimen would have been statistically unreliable. To check for major confounding with the data we had, we compared baseline demographics, tumor factors, and treatment factors between the two groups defined by the ΔSUVp cutoff (0.69); none of these differed significantly (Table 5; all p > 0.05). This comparison has been added to the manuscript as Table 5 (Lines 285-298). This supports the view that the prognostic effect of ΔSUVp was not simply due to an obvious imbalance in baseline risk. We have clarified this point in the revised Limitations section (Lines 396-402).

Issue 1-2: The heterogeneity in chemotherapy protocols (cisplatin alone vs. combination regimens) could introduce bias in metabolic response and outcomes.

Response 1-2: Thank you for this comment. We agree that variation in concurrent chemotherapy (cisplatin alone, cisplatin-based combinations, and a few carboplatin-based regimens) could act as a confounder. Because the cohort was small and non–cisplatin regimens were rare (n = 7), we could not meaningfully perform regimen-stratified survival analyses or include regimen in a multivariable model. Instead, we compared chemotherapy regimens between the two ΔSUVp groups (Table 5) and found no significant difference (p = 0.677), suggesting that the prognostic effect of ΔSUVp was not simply due to uneven allocation of more intensive chemotherapy. We have clarified this point in the revised Limitations section (Lines 400-402).

Issue 1-3: ROC-derived cutoffs may be overfitted due to the small dataset and lack of cross-validation.

Response 1-3: Thank you for this comment. We agree that ROC-based cutoff from a small cohort (n = 35) can easily overfit to the data. Because of the limited sample, we could not do meaningful internal validation, so the Youden-based ΔSUVp cutoff (0.69) should be regarded as exploratory. We have added this clarification to the Limitations and noted that it should be confirmed in larger, independent cohorts (Lines 409-417).

Issue 1-4: In the introduction or discussion part, following related references could be added to strengthen the paper: PMID: 40336569; doi.org/10.1002/pro6.70007.

Response 1-4: Thank you for this comment. We have now incorporated both references into the Discussion (Lines 356-364) to place our hypopharyngeal SCC–specific findings within the broader trend toward subsite-tailored strategies in head and neck oncology. We believe this addition clarifies that our iPET-based prognostic approach is consistent with the current movement toward subsite-specific management in head and neck cancers.

Issue 1-5: The statistical analysis relied solely on univariate Cox regression, which may exaggerate the predictive value of ΔSUVp.

Response 1-5: Thank you for this important comment. We agree that using only univariate Cox regression is a limitation and that a multivariable model would be required to prove that ΔSUVp is an independent prognostic factor. However, as we now state more clearly in the Discussion, our cohort was small (n = 35) and only 12 disease recurrences occurred, so constructing a multivariable Cox model with several clinical covariates would have led to overfitting and unstable estimates. To at least check for major confounding, we compared baseline tumor and treatment characteristics between the two groups defined by the ROC-derived ΔSUVp cutoff (0.69). No significant differences were found (Table 5), which suggests that the prognostic effect of ΔSUVp was not confounded by significant baseline differences in key baseline characteristics. We have clarified this point in the revised Limitations section (Lines 396-402). Thus, we present the current findings as hypothesis-generating and in need of confirmation in a larger cohort.

Issue 1-6: The clinical implications are limited, as how the identified thresholds would guide treatment adaptation remains speculative.

Response 1-6: Thank you for this important comment. We agree that, at present, the clinical implications of the identified thresholds remain partly speculative, because we did not test treatment adaptation (e.g., dose escalation) based on these values. As such, the proposed cutoff should be regarded as exploratory and hypothesis-generating, and its utility will need to be verified in future prospective trials.

Issue 1-7: The methodology for PET segmentation (fixed SUV = 2.5) may not be optimal for all lesions, potentially affecting MTV/TLG quantification. Could the authors give us an explanation?

Response 1-7: We appreciate this comment. We agree that a fixed SUV threshold of 2.5 is not optimal for every lesion and can make MTV/TLG partly method-dependent. In this study we adopted SUV 2.5 because this was the segmentation criterion used in previous studies and meta-analyses on PET-based volumetric parameters in head and neck cancer, and we applied the same criterion to both prePET and iPET to keep the measurements uniform.

Issue 1-8: No discussion was provided on interscan variability, attenuation correction, or scanner calibration, which could influence SUV reproducibility.

Response 1-8: Thank you for pointing this out. We agree that technical factors such as scan-to-scan variability, attenuation correction, and scanner performance are important for SUV reproducibility. To clarify how these factors were controlled in this prospective study, we have revised the Methods (PET/CT scanning technique) section to state explicitly that all pre-treatment and interim scans were performed on the same PET/CT system using an identical acquisition and reconstruction protocol, including CT-based attenuation correction (Lines 97–111). Because this was a single-center study using one scanner and one protocol, we believe that technical variability was minimized as far as possible, and that the observed changes in SUV (e.g. ΔSUVp) are more likely to reflect biological response than scanner-related artifacts.

Issue 1-9: Figures and tables could be improved for clarity, with overlapping abbreviations and dense data presentation making interpretation difficult.

Response 1-9: We thank the reviewer for these insightful comments regarding the clarity of our figure and tables. We agree that the initial presentation, with dense data and overlapping abbreviations, could hinder interpretation. To address this, we have undertaken several revisions across the Results section and all visual elements.

First, to enhance the structural clarity of our findings, we reorganized the Results section into five distinct subheadings. This systematic framework separates descriptions of PET metrics from survival analyses and thereby reduces the perceived density of the information presented. Second, we focused on the visual optimization of the figure. We increased the font size of axis labels to avoid visual crowding. Third, we thoroughly revised the tables to improve readability. We adjusted column alignment, simplified the presentation where possible, and carefully reviewed and standardized the use and definition of all abbreviations throughout the main text, figure, and tables.

We believe that these combined revisions substantially improve the clarity, readability, and overall professional quality of our complex data presentation, thereby simplifying interpretation for the reader.

Issue 1-10: Finally, the follow-up duration and censoring information are not clearly detailed for survival analyses, raising concerns about data completeness and robustness.

Response 1-10: Thank you for pointing out this issue. We agree that the rules for follow-up and censoring need to be stated explicitly to judge the robustness of the survival analyses. Our study followed all patients prospectively with regular clinical and imaging assessments, achieving a median follow-up of 52 months for the entire cohort, as shown in the Results. To address the concern regarding data completeness, we have revised the Statistical analysis section (Lines 142-146) to clearly specify the follow-up duration and the definitions for censoring. We believe this addition clarifies the completeness of follow-up and addresses the concern regarding censoring.

Rebuttal Letter

We sincerely thank the editor and reviewers for their thoughtful and constructive comments on our manuscript. We have carefully revised the manuscript in line with these suggestions. Below, we present a point-by-point response to each comment. All corresponding revisions are highlighted in the updated manuscript.

Reviewer #2: It was a pleasure to review your paper on a prospective study on the utilization of interim FDG PET/CT scanning during the chemoRT of patients with specifically hypopharyngeal cancer as a prognostic indicator. It has good data, decent design, and aligns with similar research in the field, although its more specific nature will assuredly give more credence to this implementation in this specific type of cancer. There are many things that I think should be addressed before acceptance. I have attached a PDF with comments and I'll summarize the most important points here.

Response: We thank the reviewer for these positive comments on the work. We have also carefully addressed the specific comments annotated in the attached PDF and have revised the manuscript accordingly.

Issue2-1:There is a lack of citing more recent relevant studies which are extremely similar and analogous to this work including: https://doi.org/10.1016/j.oraloncology.2018.11.005 ; https://doi.org/10.1002/lary.24826 ; https://doi.org/10.1007/s00259-017-3836-8 ; and more found in this review: https://doi.org/10.1016/j.ijrobp.2017.02.217.

Response:2-1 We thank the reviewer for pointing out this issue and for providing a comprehensive set of references. We carefully reviewed the suggested, closely related studies and appreciate your highlighting these important works. Two of the four articles were already cited and discussed in our manuscript: Chen et al. (now cited as [14]) is discussed in the Discussion (lines 332–334), and the systematic review by Garibaldi et al. (now cited as [16]) is discussed in the Discussion (lines 375–378). We agree that the other two studies, Martens et al. and Kim et al., are highly relevant and further strengthen our Discussion. As suggested, we have revised the Discussion to incorporate these citations, and the corresponding additions are located on lines 334–337.As a result of adding new references, all references have been renumbered throughout. Only the newly added text and citations are highlighted in the marked manuscript.

Issue2-2: There is a lack of an important table looking at your defined prognostic cutoff and the demographics to address confounders.

Response2-2: We thank the reviewer for these comments. To address potential confounding, we prepared a balance table stratified by the ΔSUVp cutoff of 0.69 and added it as Table 5. Between the high-risk (≤0.69) and low-risk (>0.69) groups, we compared baseline characteristics using the Mann–Whitney U test for continuous variables (age) and Fisher’s exact test for categorical variables (sex, ECOG PS, smoking history, primary tumor subsite, T/N classification, stage, radiotherapy technique (IMRT vs 3D-CRT), and chemotherapeutic regimen). All tests were two-sided with α = 0.05. As shown in Table 5, no statistically significant imbalances were observed across these covariates (all p > 0.05), suggesting that the prognostic association of ΔSUVp is unlikely to be driven by these baseline factors in our cohort. We updated the Statistical analysis subsection to describe these procedures (lines 159–164) and reference Table 5 in the Results .

Issue2-3: The supplemental raw data is missing.

Response2-3: We apologize for this oversight. We agree that data accessibility is essential for transparency. We have now uploaded the complete raw data set as S1 File in the revised submission.

Issue2-4: There is a lack of explanation about the machines and software used. There is a lack of discussion on the limitations and about the lack of blinding, as well as how treatment may have been changed relative to the ΔSUVmax or not. These things should be explained.

Response2-4: We thank the reviewer for these insightful comments. The requested details regarding the machines and software have been added to the “PET/CT scanning technique,” “FDG-PET image interpretation and metabolic parameter measurement,” and “Statistical analysis” subsections in the Materials and Methods section. Specifically, the PET/CT scanning technique subsection identifies the scanner as the GE Discovery PET/CT 690 (GE Healthcare, Milwaukee, WI). The PET/CT image interpretation and metabolic parameter measurement subsection notes that metrics were extracted using SYNAPSE SAI Viewer v2.0 (FUJIFILM, Tokyo, Japan). Finally, the Statistical analysis subsection indicates that analyses were performed with IBM SPSS Statistics v23 (IBM Corp., Armonk, NY).

We agree that the limitations paragraph would be significantly strengthened by explicitly addressing the lack of blinding and clarifying whether interim FDG PET/CT (iPET) results led to treatment modifications. This was a prospective observational cohort study, and the results (including ΔSUVp) were not used to make adaptive treatment decisions. The treating physicians were not blinded to the iPET results, but the treatment was completed as planned in all cases. This clarifies that our findings on prognosis were not confounded by iPET-based treatment modifications.

We also agree that the lack of blinding during image analysis is a limitation and we have revised the limitations paragraph in the Discussion section to explicitly include these points (lines 403–408).

Issue2-5: You define a cutoff by maximizing Younden via the ROC, but this may not be the best approach, there should be discussion about this and the utility for prognostication for higher sensitivity or specificity.

Response2-5: We thank the reviewer for these insightful comments. We fully agree that the optimal cutoff should be tailored to the intended clinical application. In clinical scenarios that prioritize the early detection of treatment-resistant disease, a sensitivity-oriented cutoff rather than a Youden-based cutoff may be more appropriate. The Youden index, while statistically balanced, may not be the most clinically useful choice in all settings. In the present study, we used the Youden index to derive an objective, ex

---

## [Decision Letter · Decision Letter 1]

4 Jan 2026

Prognostic role of interim F-18 fluorodeoxyglucose positron emission tomography-computed tomography during chemoradiation therapy in patients with hypopharyngeal squamous cell carcinoma

PONE-D-25-48920R1

Dear Dr. Nomoto,

We’re pleased to inform you that your manuscript has been judged scientifically suitable for publication and will be formally accepted for publication once it meets all outstanding technical requirements.

Kind regards,

Carmelo Caldarella, Ph.D., M.D.

Academic Editor

PLOS One

Additional Editor Comments (optional):

Reviewers' comments:

Reviewer's Responses to Questions

**Comments to the Author**

Reviewer #4: All comments have been addressed

Reviewer #5: All comments have been addressed

2. Is the manuscript technically sound, and do the data support the conclusions?

Reviewer #4: Yes

Reviewer #5: Yes

3. Has the statistical analysis been performed appropriately and rigorously?

Reviewer #4: Yes

Reviewer #5: (No Response)

4. Have the authors made all data underlying the findings in their manuscript fully available?

Reviewer #4: Yes

Reviewer #5: Yes

5. Is the manuscript presented in an intelligible fashion and written in standard English?

Reviewer #4: Yes

Reviewer #5: Yes

Reviewer #4: (No Response)

Reviewer #5: Authors submitted al requested changes. No further changes are reqired. In my opinion this manuscript is adequate for publication in this present form.

**Do you want your identity to be public for this peer review?** For information about this choice, including consent withdrawal, please see our Privacy Policy

Reviewer #4: No

Reviewer #5: No

---

## [Editor Report · Acceptance letter]

PONE-D-25-48920R1

PLOS One

Dear Dr. Nomoto,

I'm pleased to inform you that your manuscript has been deemed suitable for publication in PLOS One. Congratulations! Your manuscript is now being handed over to our production team.

Kind regards,

on behalf of

Dr. Carmelo Caldarella

Academic Editor

PLOS One